# Introgression of Heterotic Genomic Segments from *Brassica carinata* into *Brassica juncea* for Enhancing Productivity

**DOI:** 10.3390/plants12081677

**Published:** 2023-04-17

**Authors:** Prashant Vasisth, Naveen Singh, Omkar Maharudra Limbalkar, Mohit Sharma, Gokulan Dhanasekaran, Mohan Lal Meena, Priyanka Jain, Sarika Jaiswal, Mir Asif Iquebal, Anshul Watts, Kiran B. Gaikwad, Rajendra Singh

**Affiliations:** 1Division of Genetics, Indian Council of Agricultural Research-Indian Agricultural Research Institute, New Delhi 110012, India; 2Division of Agricultural Bioinformatics, Indian Council of Agricultural Research-Indian Agricultural Statistics Research Institute, New Delhi 110012, India; 3Indian Council of Agricultural Research-National Institute of Plant Biotechnology, New Delhi 110012, India

**Keywords:** *B. carinata*, *B. juncea*, interspecific hybridization, heterosis, heterotic genomic segments, introgression lines

## Abstract

Interspecific hybridization resulted in the creation of *B. juncea* introgression lines (ILs) generated from *B. carinata* with increased productivity and adaptability. Forty ILs were crossed with their respective *B. juncea* recipient parents to generate introgression line hybrids (ILHs) and the common tester (SEJ 8) was used to generate test hybrids (THs). Mid-parent heterosis in ILHs and standard heterosis in THs were calculated for eight yield and yield-related traits. Heterotic genomic regions were dissected using ten ILs with significant mid-parent heterosis in ILHs and standard heterosis in THs for seed yield. A high level of heterosis for seed yield was contributed by 1000 seed weight (13.48%) in D31_ILHs and by total siliquae/plant (14.01%) and siliqua length (10.56%) in PM30_ILHs. The heterotic ILs of DRMRIJ 31 and Pusa Mustard 30 were examined using polymorphic SNPs between the parents, and a total of 254 and 335 introgressed heterotic segments were identified, respectively. This investigation discovered potential genes, viz., *PUB10*, *glutathione S transferase*, *TT4*, *SGT*, *FLA3*, *AP2/ERF*, *SANT4*, *MYB*, and *UDP-glucosyl transferase 73B3* that were previously reported to regulate yield-related traits. The heterozygosity of the *FLA3* gene significantly improved siliqua length and seeds per siliqua in ILHs of Pusa Mustard 30. This research proved that interspecific hybridization is an effective means of increasing the diversity of cultivated species by introducing new genetic variants and improving the level of heterosis.

## 1. Introduction

The rapeseed-mustard group comprises six agriculturally important crops, out of which *B. rapa* (2n = 20, AA), *B. nigra* (2n = 16, BB), and *B. oleracea* (2n = 18, CC) are diploids; whereas, *B. juncea* (2n = 36, AABB), *B. napus* (2n = 38, AACC), and *B. carinata* (2n = 34, BBCC) are digenomic tetraploids, which evolved in nature following hybridization between the constituent diploid species [1]. It is the second most important group of oilseed crops in the world after soybean, with a production of 73.89 million metric tonnes, and contributed 12.16% to global oilseed production (607.50 million metric tons) in 2020–2021 [2]. India is the third-largest producer of rapeseed-mustard, followed by Canada and China, with a total production of 8.5 million metric tonnes from a 6.70 mha area [2].

Among the different species of the genus *Brassica*, *B. juncea* (Indian mustard) is predominantly cultivated in India and accounts for more than 90% of its acreage [3,4]. With consistent efforts, highly productive varieties of oilseed crops have been developed and commercialized in the country. Despite the availability of improved cultivars, the import of edible oil is increasing year after year due to population growth and changing lifestyles [4]. The current production of edible oil is 11 million tonnes, but we need more than 25 million tonnes in order to achieve national self-sufficiency in edible oils [4]. *B. juncea*, being a pivotal crop in the edible oil sector, needs to be genetically improved for productivity traits to meet the edible oil requirements and, hence, the improvement of seed yield and oil content in *B. juncea* remains a major priority. Current varietal development programmes are ineffective in increasing productivity to the desired level [5]. Hybrids have contributed significantly to food security by increasing yield per unit area worldwide in many agriculturally important crops [6]. Therefore, breeding for higher heterosis could play an important role in the improvement of crop productivity, nutrient quality, and resistance to biotic and abiotic stresses [7]. 

*B. juncea* has a narrow gene pool because more than 90% of present-day varieties in India are either derived directly from the cultivar Varuna or its derivatives [3]. A narrow genetic base is a major problem in the exploitation of heterosis as the level of heterosis increases with the increase in genetic diversity between parents and, hence, interspecific hybrids were found to be more promising than intraspecific hybrids [8]. To exploit the hybrid vigour in *B. juncea*, the genetic base of the working germplasm needs to be broadened. Interspecific hybridization is a powerful tool for targeted crop improvement, broadening the genetic base of a species, introgression of favorable adaptive alleles between species, and the exploitation of heterosis [9,10]. It brings about large-scale random introgression of alien segments, which ultimately broadens the genetic base of the species [11]. Efforts have been made to utilize the primary and secondary gene pool to broaden the genetic base in different species. Udall et al. [12] developed hybrids between introgression lines (ILs) of *B. napus* having introgression segments from Chinese semi-winter type with natural spring *B. napus* and reported heterosis for seed yield in hybrids. Qian et al. [13] concluded that Chinese *B. rapa* can be a valuable gene pool for increasing biomass in *B. napus*. Interspecific hybridization between *B. juncea* and *B. carinata* holds promise for introgressing desirable traits from one species to another. Inter-subgenomic heterosis was also observed when *B. rapa* and *B. carinata*-derived *B. juncea* introgression lines were crossed with *B. juncea* genotypes [14]. Moreover, *B. carinata* (BBCC) is endowed with many agronomically desirable traits such as resistance/tolerance to abiotic stresses (drought and heat); and biotic stresses, viz., aphid, Sclerotinia rot, white rust, Alternaria blight, blackleg, and powdery mildew [15]. Successful attempts of interspecific hybridization were made to transfer favourable alleles from *B. carinata* to *B. rapa* [16,17], *B. napus* [18], and *B. juncea* [7,14,19,20,21]. Nikzad et al. [22] used *B. oleracea* to exploit heterosis in hybrids between *B. napus* and *B. oleracea*; and suggested that the C genome of *B. oleracea* has potential to improve the seed yield in hybrid cultivars.

Therefore, interspecific hybridization was used at the Indian Council of Agricultural Research-Indian Agricultural Research Institute (ICAR-IARI), New Delhi to enrich the genome of *B. juncea* with favorable adaptive alleles of *B. carinata*, resulting in the development of morphologically and cytogenetically stable *B. carinata*-derived *B. juncea* ILs (2n = 18II). Using these lines, we have successfully demonstrated higher yield heterosis and drought tolerance in a set of 105 hybrids [23]. These outcomes have further encouraged us to identify heterotic genomic segments and dissect the responsive genes in *B. carinata*-derived *B. juncea* ILs. Hence, the present study ensures the first use of ILs derived from the interspecific cross between *B. carinata* accession BC-4 and *B. juncea* genotypes (DRMRIJ 31 and Pusa Mustard 30) for the identification of heterotic genomic segments and underlying candidate genes introgressed from *B. carinata* accession BC-4 to *B. juncea* genotypes.

## 2. Results

### 2.1. Mean Performance of Introgression Lines and Hybrids

Observations on ILs, ILHs, and THs for yield and yield-contributing traits were recorded for both the genetic backgrounds. An analysis of variance revealed the significance of the mean sum of squares due to treatments for all of the traits, except for harvest index in test hybrids generated in the genetic background of Pusa Mustard 30 (Appendix A). Both sets of ILs in the background of DRMRIJ 31 (D31_ILs) and Pusa Mustard 30 (PM30_ILs) exhibited a wide range of phenotypic variability for all the traits studied (Table 1). The superior performance of ILHs and THs in comparison to ILs was observed for most of the traits. When DRMRIJ 31-derived ILs were used in generating hybrids, their THs were phenotypically superior to ILHs, and ILHs were superior to ILs for all the traits except 1000 seed weight (g). Oil content (%) mean values were higher in THs than in ILHs and ILs in both the genetic backgrounds. In the backgrounds of DRMRIJ 31 and Pusa Mustard 30, ILHs were observed to be superior to ILs and THs for 1000 seed weight (g). Furthermore, in their respective genetic backgrounds, the mean seed yields (t/ha) of THs and ILHs were significantly higher than those of ILs.

Seed yield in PM30_ILs ranged from 1.77–2.83 t/ha with an average of 2.38 t/ha, and the mean seed yield of PM30_ILs and PM30_ILHs was found to be higher than the seed yield of Pusa Mustard 30 (2.03 t/ha). Similar results were observed for mean seed yield (t/ha) in the genetic background of DRMRIJ 31. Total siliquae per plant and 1000 seed weight exhibited higher values for D31_ILs than the recipient parent DRMRIJ 31. Similarly, PM30_ILs were superior to Pusa Mustard 30 for seeds per siliqua, total siliquae on main shoot, total siliquae per plant, oil content, and harvest index (Table 1). In general, the mean seed yield (t/ha) of test hybrids was significantly higher than the values of their respective check hybrids in all the genetic backgrounds. Further, a wide range of variations were observed for all the traits in the test hybrids (Appendix A).

### 2.2. Heterosis Expressed in Introgression Line Hybrids and Test Hybrids

Mid-parent heterosis was calculated for introgression line hybrids (ILHs) generated by crossing ILs with their respective recipient parent; and for the set of test hybrids created by crossing ILs with standard tester SEJ 8, standard heterosis was calculated over respective check hybrids. Recipient parents, viz., DRMRIJ 31 and Pusa Mustard 30 were crossed with SEJ 8 to generate check hybrids (CHs). Wide ranges for mid-parent heterosis over introgression lines and respective genetic backgrounds, and standard heterosis over respective check hybrids were observed (Table 2 and Table 3). The highest mid-parent heterosis for seed yield was observed for D31_ILHs (76.81%), followed by PM30_ILHs (58.40%). D31_THs had the highest standard heterosis (42.67%) over respective check hybrids for seed yield (t/ha), then PM30_THs (18.29%). Seed yield (t/ha) exhibited the same pattern in both sets of ILHs with the highest mid-parent heterosis. All the traits recorded positive mid-parent heterosis in both sets of ILHs, except total siliquae on the main shoot (−2.73%) in the background of DRMRIJ 31. Higher standard heterosis was reported in THs, over respective check hybrids, for traits like total siliquae on the main shoot, total siliquae/plant and seed yield (t/ha) (Table 3).

### 2.3. Confirmation of Heterosis Arising due to Introgressed Segments 

A significant difference between the mid-parental value and ILHs or THs and CHs indicates the presence of heterotic loci in the parental line(s). In the present investigation, both experiments were conducted to establish the presence of heterotic genomic segments in the *B. carinata*-derived *B. juncea* introgression lines. For all the traits, hybrids with significant mid-parent heterosis and standard heterosis over the respective check hybrids are presented in Appendix A. The highest number of hybrids with significant mid-parent heterosis was reported for seed yield (30), followed by siliqua length (17), and thousand seed weight (14); whereas, the maximum number of hybrids that expressed standard heterosis over their respective check hybrids was observed for total siliquae on the main shoot (22), followed by total siliquae/plant (19), and seed yield (17). The best 10 ILHs (5 from each genetic background), selected on the basis of higher mid-parent heterosis for seed yield (Table 4), were compared with their respective parents for seed yield (Figure 1a,b) to access the effect of introgression segments in homozygous and heterozygous conditions. THs generated using introgression lines, also used in generating ILHs, with the same genetic background were then compared to their respective check hybrids. Despite the inferior performance of a few ILs when compared to their respective genetic backgrounds, their derived ILHs were found to be superior to parents, ILs and their respective genetic backgrounds. All D31_ILs, except D31_IL136, were identified as inferior to DRMRIJ 31, whereas D31_ILHs derived from them were significantly superior to both parents. All of the selected ILHs developed in both genetic backgrounds had higher seed yields (t/ha) when compared to their parents. To validate the expression of heterosis by ILs in ILHs, 10 THs were generated using these heterotic ILs as one parent and SEJ 8 as another. A comparison of THs with their respective CHs (genetic backgrounds × SEJ 8) is presented in Figure 2a,b. All 10 THs (5 D31_THs and 5 PM30_THs) were found to be superior to their respective CHs in terms of seed yield (t/ha), with the exception of TH; PM30_IL180 × SEJ 8, which yielded less than their respective CH (Pusa Mustard 30 × SEJ 8).

### 2.4. Marker Analysis and Detection of Introgressed B. carinata Genomic Segments

Genotyping by Sequencing (GBS) was used for the identification of Single Nucleotide Polymorphisms (SNPs) between introgression lines and their donor parent (*Brassica carinata* Acc. BC-4). Polymorphic 5157 SNPs between DRMRIJ 31 and 4893 between Pusa Mustard 30 and BC-4 were identified. These polymorphic markers were used to genotype introgression lines in order to identify genomic segments introgressed from alien donor species and responsible for improved hybrid performance. Graphical genotypes for each introgression line were generated using the Graphical GenoType 2.0 software (Figure 3a,b). Introgression lines derived from the genetic background of DRMRIJ 31 carry 3.1% of the genome from *B. carinata* and 57.3% of it from *B. juncea* (DRMRIJ 31), with an average heterozygosity of 5%. Among these lines, D31_IL136 had the highest number of introgressed donor genome segments (169), followed by D31_IL94 (109), and the maximum number of introgressed segments (117) were observed on the B07 chromosome (Appendix A). In ILs derived from the genetic background of Pusa Mustard 30, an average of 5.7% of the genome was from *B. carinata* and 64.3% from *B. juncea* (Pusa Mustard 30), with an observed heterozygosity of 6.2%. PM30_IL177 (108) had the most introgressed segments, followed by PM30_IL181 (99), and chromosome B08 had higher introgressed segments as compared to other chromosomes (Appendix A). Overall, the “B” sub-genome carries a higher number of introgressed segments than that of the “A” sub-genome among the ILs developed in both genetic backgrounds (Figure 4). Despite a lower percentage of *B. carinata* introgression, a higher number of smaller segments were reported in introgression lines. These segments were also linked to expressed heterosis in hybrids derived from them, as well as the discovery of candidate genes within these heterotic segments.

### 2.5. Identification of Heterotic Genomic Segments in ILs and Associated Candidate Genes

Ten ILHs, generated by five ILs each from both genetic backgrounds, were used to study the contribution of alien introgression segments to the expression of heterosis in hybrids. The traits responsible for the superior performance of hybrids were different in hybrids developed from ILs and their respective genetic backgrounds.

The superior performance of hybrids D31_ILH89, D31_ILH105, D31_ILH120, D31_ILH136, and D31_ILH156—generated by involving ILs of DRMRIJ 31 genetic background (D31_ILs)—was attributed to improved 1000 seed weight (g) and harvest index (%). Additionally, these hybrids were also superior to their respective D31_ILs for 1000 seed weight (g) and harvest index (%) (Table 4). D31_IL136 has 169, D31_IL89 has 16, D31_IL105 has 14, D31_120 has 6, and D31_IL156 has 49 introgressed segments from *B. carinata* acc. BC-4 (Appendix A). Identified SNPs in introgressed segments of donor parent BC-4 could thus be associated with increased 1000 seed weight (g) (Figure 5) and harvest index (%) in D31_ILHs. A total of 273 candidate genes were identified in the heterotic D31_ILs (Appendix A), and a few of them were found common among the D31_ILs. Candidate genes, viz., glutathione S transferase, *MYB 94*, *BRI 1*, *TT4*, *SGT*, *FT*, *EMB*, and *ARFs* found to control 1000 seed weight (g) in D31_ILs, were also reported in previous studies to regulate seed size (Appendix A).

Higher mean performance in hybrids, viz., PM30_ILH168, PM30_ILH180, PM30_ILH185, PM30_ILH187, and PM30_ILH190 was due to the increase in total siliquae per plant and oil content (%); whereas hybrids, viz., PM30_ILH180, PM30_ILH187, and PM30_ILH190 outperformed by increasing siliquae length (cm) and the number of seeds per siliqua (Table 4, Figure 6a,b). A total of 352 candidate genes were identified in heterotic PM30_ILs, and these could be associated with the expression of heterosis in PM30_ILHs (Appendix A). The *FLA3* gene (fasciclin-like arabinogalactan protein 3 precursor) was identified in most of the heterotic PM30_ILs. This gene was previously reported to be involved in the regulation of siliquae development and siliqua length, and is expected to contribute to the superior performance of PM30_ILHs in this study (Appendix A). 

Furthermore, candidate genes, viz., *AP2/ERF*, *SANT4*, *MYB*, zinc finger domain, Protein kinases, glutathione s transferase, and homeobox domains were common among heterotic ILs developed in both the genetic backgrounds (Appendix A).

## 3. Discussion

Interspecific hybridization proved to be the most prominent approach for expanding the genetic variation in cultivated cultivars of diploid and polyploid species [10,24]. Among *Brassicas*, *B. carinata* is well suited for broadening the genetic base of other *Brassicas* since it is highly resistant to biotic [20] and abiotic [14,23] stresses. Genetic background plays an important role in the expression of heterosis [25]; therefore, the development of ILs in improved genetic backgrounds is desired. The present study was thus directed towards improving the genetic base of *B. juncea* through interspecific hybridization with *B. carinata,* by developing *B. carinata*-derived *B. juncea* introgression lines (ILs) in the genetic backgrounds of cultivars DRMRIJ 31 and Pusa Mustard 30. Wei et al. [14] demonstrated that the integration of *B. carinata* and *B. rapa* sub-genomes into the *B. juncea* genome resulted in improvements for thousand seed weight and seeds per siliqua, but the performance of these traits in the new *B. juncea* remained inferior. However, inter-subgenomic hybrids between the introgressed *Brassica* line and the recipient *B. juncea* parent showed considerable potential for heterosis. 

The wide variation observed among ILs for all traits reported in this study, as compared to their respective genetic backgrounds, suggests that interspecific hybridization can be useful for increasing variability among ILs (Table 1 and Appendix A). The phenotypic values of almost all the traits studied were higher in introgression lines than in their respective genetic backgrounds, indicating that introgressed segments from *B. carinata* caused desirable trait-specific variations among ILs. Results suggested that the use of biparental mating among F_2_ plants and selfing thereafter has potentially increased the variation among introgression lines for yield-related traits. Moreover, Schelfhout et al. [26] reported that interspecific hybridization between *B. juncea* and *B. napus* followed by selfing led to enhanced genetic diversity in canola quality progeny of both species without loss of donor alleles, whereas repeated backcrossing generally associated with loss of donor parent alleles. Hybrids generated between ILs and their respective genetic backgrounds (ILHs) expressed higher heterosis for yield and its contributing traits. The superiority of introgression line hybrids as compared to introgression lines indicates that heterotic segments (heterozygous introgressed segments) were responsible for the expression of this heterosis [27]. Gupta et al. [25] developed C-chromosome substituted lines in *B. juncea* backgrounds that had higher main shoot length and total siliqua on the main shoot as compared to normal *B. juncea* genotypes. Further, higher heterosis was reported in hybrids generated between these chromosomal substitution lines and the original lines. In another recent effort, Zhang et al. [24] expanded the genetic diversity in *B. juncea* through the introgression of the *B. rapa* genome.

### 3.1. Dissection of Heterotic Genomic Segments

Studies on heterosis require a different kind of population as compared to conventional QTL mapping. The population required for the identification of heterotic loci and heterotic effects should be derived from elite hybrids with a strong heterozygous advantage over parents [28]. Heterotic loci and heterotic effects can be identified by the F_2_ [29], immortalized F_2_ and DH [30], RILBC_1_ [31], and the CSSLs/ILs [32,33,34,35,36] populations. Due to the low background noise caused by the presence of only a few introgressed segments, ILs are valuable resources for identifying heterotic loci, heterotic effects, and desirable allelic interactions [37,38]. Therefore, *B. carinata*-derived *B. juncea* introgression lines were used in the present study for the identification of heterotic genomic segments.

In general, the average number of *B. carinata* introgressed segments was higher and the percent introgression was lower, indicating that the size of the introgressed segments was small but the number of introgressed segments was quite high in each IL. This is primarily due to biparental mating in the F_2_ generation followed by selection in each filial generation for the development of ILs, which led to the breakage of undesirable linkage and reduced the size of introgressed segments. Quezada-Martinez et al. [39] also suggested that introgressed large genomic segments with undesirable effects could be reduced into smaller segments through subsequent recombination using backcrossing or biparental mating. A high number of introgressed segments were due to pairing and recombination between “B” sub-genomes of *B. carinata* and *B. juncea*, and higher chances of recombination between A and C sub-genomes [40,41]. Introgressed segments were reported significantly higher in “B” than “A” sub-genome of *B. juncea.* Less introgression into “A” sub-genome of *B. juncea* is, therefore, anticipated due to directional phenotypic selection towards *B. juncea*. “A” sub-genome of *B. juncea* contributes maximally to the seed yield [42], and any major change to the “A” sub-genome is expected to reduce its performance. Therefore, lower performing segregants with higher degrees of introgression were rejected in the process of development of ILs. 

Introgression lines were developed in two genetic backgrounds (DRMRIJ 31 and Pusa Mustard 30) and hybridized with respective genetic backgrounds to generate Introgression Line Hybrids (ILHs), and with a common tester to generate Test Hybrids (THs). Previous studies have identified heterotic loci on the basis of mid-parent heterosis (33–35) or heterosis over check hybrids in test cross populations [36]. However, we included both backcross and testcross populations in this study, and heterotic segments were dissected from ILs exhibiting higher heterosis in both ILHs and THs. SNP markers (polymorphic between *B. carinata* accession BC-4 and respective genetic backgrounds) were used to identify 254 and 335 heterotic genomic segments (Appendix A) from D31_ILs and PM30_ILs, respectively, and representative markers from heterotic segments were used to identify candidate genes. This approach to the dissection of heterotic segments was in close accordance with the studies conducted by Gaikwad et al. [9] and Nassirou et al. [32]. Introgressed segments from the donor parent (BC-4) could explain the high heterosis in yield and yield-related traits in this study. Although heterotic loci for yield-influencing traits in two backcross populations developed using double haploid lines were identified [43], the identification of heterotic segments in *B. juncea* using introgression lines has not been reported so far. Similar marker introgression was also identified in rice using introgression lines, and these marker introgressions were associated with the traits that contributed to the heterosis [9].

### 3.2. Candidate Genes Associated with Heterotic Genomic Segments

Heterotic genomic segments, introgressed from *B. carinata* accession BC-4 into the background of *B. juncea* were identified, and representative markers from each segment were used for candidate gene analysis. The flanking sequences to the polymorphic SNPs were determined using BLAST with the reference genome, and 1000 base pairs upstream and downstream to the position of the SNPs were taken for annotation with the *Arabidopsis* genome for identification of candidate genes. To identify candidate genes, a similar approach was followed by Cai et al. [44] and Wang et al. [45] in *B. napus*. Two hundred and seventy-three candidate genes in heterotic segments of D31_ILs and 352 in heterotic segments of PM30_ILs were observed (Appendix A). The 1000 seed weight contributed significantly to the higher heterosis for seed yield in D31_ILs (Table 4). Seed size is regulated by differential cell proliferation and cell expansion in different compartments of the seed, which affects the final seed size. Furthermore, seed size is regulated by various transcription factor families and the interplay of plant hormones, viz., auxin, cytokinin, ethylene, and brassinosteroids [46]. In heterotic D31_ILs, transcription factors, viz., *WRKY*, *MYB*, *ARF*, *TCP*, *BRI1*, *UBP 26*, *TT4*, *FT*-interacting protein, and genes related to cell wall biogenesis, such as expansin, extension, and PME were observed (Appendix A, Figure 5). These candidate genes were already reported to regulate seed size in the genus *Brassica* [46,47,48,49,50]. In the case of heterotic PM30_ILs, siliqua length was found to be a major contributor, along with oil content, to the higher heterosis for seed yield. Candidate gene *FLA3*, identified in heterotic PM30_ILs, was responsible for increased siliqua length and seeds per siliqua in PM30_ILHs (Figure 6a,b). *FLA3* gene, which regulates the elongation of the stamen filament and female fertility in *Arabidopsis*, is already known to influence siliqua length [45,51]. Candidate genes, viz., seed storage albumin 1 (*SESA1*), *SANT 4*, *CYP72A13*, zinc finger/ring type, Glycoside hydrolase family 9, potassium transporter, UDP-glucosyl transferase 73B3, and 3-ketoacyl-acyl carrier protein synthase III were reported in heterotic PM30_ILs (Appendix A). These candidate genes were reported to have an association with oil content in *B. juncea* [52]. Common candidate genes, viz., *AP2/ERF*, *SANT4*, *MYB*, zinc finger domain, Protein kinases, glutathione s transferase, and homeobox domains were observed among heterotic ILs developed in different genetic backgrounds (Appendix A). Previous studies have confirmed that these candidate genes have an important role in the regulation of yield-related traits [52,53,54]. These candidate genes can be deployed in parental lines through marker-assisted selection to improve yield-related traits in hybrids after identifying suitable molecular markers.

## 4. Materials and Methods

### 4.1. Plant Materials

Interspecific hybridization between *B*. *juncea* genotypes, viz., DRMRIJ 31 and Pusa Mustard 30, and a *B*. *carinata* accession (BC-4) followed by biparental mating in the F_2_ generation, and continuous selection and selfing thereafter, has led to the development of *B*. *carinata*-derived *B*. *juncea* introgression lines (ILs). Morphological resemblance to the respective *B*. *juncea* genotype was preferred while selecting single plants in segregating generations along with valuable transgressive segregants. This effort has resulted in the development of more than 200 lines. From this set, we included 40 ILs carrying *B. carinata* genomic segments in the genetic background of two *B. juncea* genotypes, viz., DRMRIJ 31 (27; D31_ILs) and Pusa Mustard 30 (13; PM30_ILs) for the present study. Using these lines, 40 Introgression Line Hybrids (ILHs) were generated by hybridizing the above-mentioned ILs and their respective genetic backgrounds. SEJ 8, a synthetic *B. juncea* genotype already known as a good combiner, was used as a tester to hybridize with all the 40 ILs to generate Test Hybrids (THs) and two Check Hybrids (CHs), viz., DRMRIJ 31x SEJ 8 and Pusa Mustard 30 × SEJ 8, thus, generating a total of 40 THs and two CHs. Hybrids were generated and their parents were selfed during the *rabi* 2020–2021 season at the Experimental Field of the Division of Genetics, ICAR-IARI, New Delhi. 

### 4.2. Phenotypic Evaluation 

Two different experiments were designed to evaluate the ILs, ILHs, THs, CHs, and their respective parents. In the first experiment, 40 ILHs, 40 ILs, and their respective genetic backgrounds were evaluated for the expression of heterosis, if any, and the identification of genomic segments responsible for its expression. To validate the reported heterosis and establish it again in a different genetic background, 40 THs and two CHs were compared in the second experiment. Both the experiments were conducted in a randomised complete block design (RCBD) with three replications at the experimental plot of Genetics Division, ICAR-IARI, New Delhi in the *rabi* 2021–2022 season. Each entry consisted of two rows of 2.5 m length (average 40 plants/plot) with a row-to-row and plant-to-plant spacing of 45 and 15 cm, respectively. Recommended agronomical practices and plant protection measures were adopted for raising a healthy crop.

Observations were recorded on nine seed yield and yield-related traits, viz., total siliquae on the main shoot, total siliquae per plant, siliqua length (cm), seeds per siliqua, seed yield per plant (g), biological yield per plant (g), seed yield (t/ha), 1000 seed weight (g), and oil content (%). Seed yield per plant and biological yield per plant were used to calculate harvest index using a standard formula (seed yield per plant/biological yield per plant × 100). Observations were recorded on five randomly selected competitive plants and averaged from every plot in each replication, except for seed yield (t/ha), where observations were recorded on the plot basis. Mean phenotypic values for all the studied traits were computed from three replications and further used for statistical analysis. 

An analysis of variance for randomized complete block design was performed for all the studied traits using the method of Panse and Sukhatme [55]. The following mathematical model was used for the RCBD design,
Yij = m + gi + rj + eij
where, Yij = Phenotypic observation of ith genotype in jth replication, m = General mean, gi = Effect of ith genotype, rj = Effect of jth replication, eij = Random error.

Mid-Parent Heterosis (MPH) and Standard Heterosis (SH) were calculated using the standard equation, MPH (%) = (ILH—MP)/MP × 100, and SH (%) = (TH—CH)/CH × 100, where MP is mid-parental value, TH is the value of test hybrid and CH is the value of check hybrid [56]. The significance of heterosis was computed at 5% probability level.

### 4.3. Genotyping by Sequencing 

DNA was isolated from young leaves of ILs and their parents following the CTAB (Cetyl Trimethyl Ammonium Bromide) modified method suggested by Saghai-Maroof et al. [57] and used for Genotyping by Sequencing [58]. The genomic services were outsourced to NxGenBio Life Sciences Private Limited, and sequence RAW data was generated for the optimized GBS library using Illumina True Seq sequencing. Sequenced reads were processed and aligned to the reference genome of *B. juncea* var. varuna UDSC Var 1.1 [59]. The perfectly matched and aligned sequences were analysed for SNP identification using the TASSELGBS analysis pipeline [60]. Polymorphic SNPs between donor parent and respective genetic backgrounds were identified. These polymorphic SNPs were used for the construction of graphical genotypes and the calculation of percent donor genome in each IL using the GGT 2.0 software [61]. The software analysed the data in such a way that chromosomal segments flanked by donor parent allele were considered as 100% donor segment, and chromosomal segments flanked by recipient parent allele were considered as 100% recipient segment. Further, chromosomal segments flanked by donor parent allele on one side and recipient parent allele on other side were considered as recombinant [62].

### 4.4. Identification of Heterotic Genomic Segments and Candidate Gene Analysis

It is assumed here that heterosis arises due to the effect of heterozygous loci; therefore, loci giving significant differences between hybrids (heterozygote) and the mean of two parents (homozygotes) were considered to be heterotic [63]. Significant differences between introgression line hybrids and their respective mid-parental values, thus, indicates the presence of heterotic genomic segments introgressed from the donor species [9,63,64,65]. Significant differences between test and check hybrids, on the other hand, also indicate the presence of heterotic genomic segments [36]. Genomic segment(s) in the developed ILs, introgressed from the *B. carinata* accession BC-4, expressing both significant mid-parent heterosis for seed yield in ILHs and significant standard heterosis for seed yield in THs over respective CHs were considered heterotic. Sequence flanking to SNPs, 1000bp upstream and downstream, from these heterotic genomic segments were mapped to *B. juncea* reference genome using the BLAST tool (https://blast.ncbi.nlm.nih.gov/Blast.cgi; accessed on 1 August 2022). Candidate genes were identified using the annotation of the *Arabidopsis* genome and physical positions of SNPs in the reference *B. juncea* genome [59].

## 5. Conclusions

Introgression of genomic segments from *B. carinata* has generated sufficient genetic variation in *B. juncea* accessions DRMRIJ 31 and Pusa Mustard 30, which in some cases do not provide any significant gain in ILs over the *B. juncea* genetic backgrounds intended to improve. Significant positive heterosis for seed yield (t/ha), expressed in ILHs and THs, indicates that homozygous introgressed segments from *B. carinata* after attaining heterozygosity in hybrids expressed higher heterosis. Marker analysis of ILs, on the other hand, also confirmed the presence of *B. carinata* segments, and the superiority or inferiority of these ILs to their respective genetic backgrounds could be attributed to introgressed segments. Introgression lines expressing higher heterosis for seed yield in ILHs and THs enabled the identification of heterotic segments and their association with yield-related traits. Candidate gene analysis revealed the presence of important transcription factors and genes which were reported to regulate critical yield related traits, viz., thousand seed weight (g), siliqua length (cm), seeds per siliqua, and oil content (%). The present study justifies the importance of interspecific hybridization in generating novel genetic variability, and its usefulness in improving the level of heterosis. Heterotic ILs, identified in this study, can be converted into cytoplasmic male sterile lines and/or fertility restorer lines for the development of commercial hybrids. Furthermore, candidate genes are envisaged for validation for their precise role in the regulation of yield and yield-related traits before their deployment through MAS or genomic selection.

## Figures and Tables

**Figure 1 plants-12-01677-f001:**
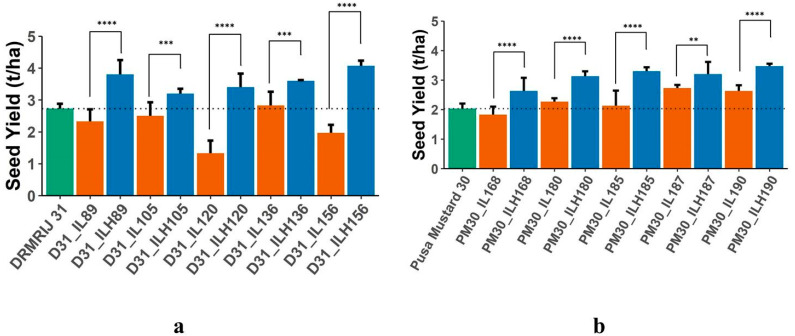
(**a**) Mean performance of DRMRIJ 31, introgression lines in its genetic background (D31_ILs) and introgression line hybrids (D31_ILHs) for seed yield; (**b**) Mean performance of Pusa Mustard 30, introgression lines in its genetic background (PM30_ILs) and introgression line hybrids (PM30_ILHs) for seed yield (** significant at *p* = 0.05; *** significant at *p* = 0.01; **** significant at *p* = 0.001).

**Figure 2 plants-12-01677-f002:**
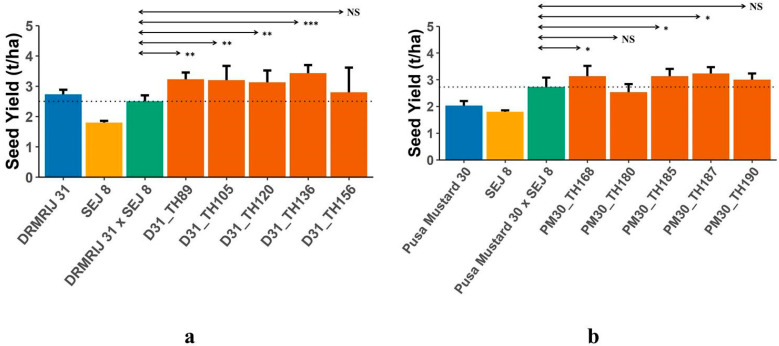
(**a**) Mean seed yield of test hybrids (D31_THs) developed using introgression lines of DRMRIJ 31 (D31_ILs) and common tester (SEJ 8) in comparison to check hybrid (DRMRIJ 31 × SEJ 8) and parents; (**b**) Mean seed yield of test hybrids (PM30_THs) developed using introgression lines of Pusa Mustard 30 (PM30_ILs) and common tester (SEJ 8) in comparison with check hybrid (Pusa Mustard 30 × SEJ 8) and parents (* significant at *p* = 0.1; ** significant at *p* = 0.05; *** significant at *p* = 0.01; NS = Non Significant).

**Figure 3 plants-12-01677-f003:**
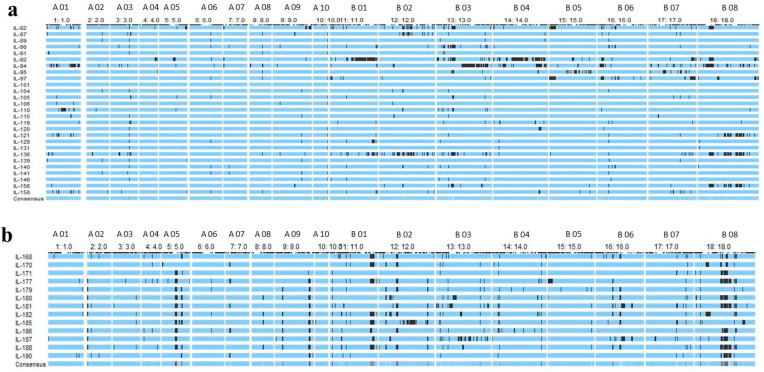
(**a**) Graphical representation of 27 ILs of DRMRIJ 31 showing proportion of donor genome (BC-4) in black colour (**b**) Graphical representation of 13 ILs of Pusa Mustard 30 showing proportion of donor genome (BC-4) in black colour (X axis represents chromosome wise distribution of introgressed segments and Y axis represents graphical genotyping of each IL).

**Figure 4 plants-12-01677-f004:**
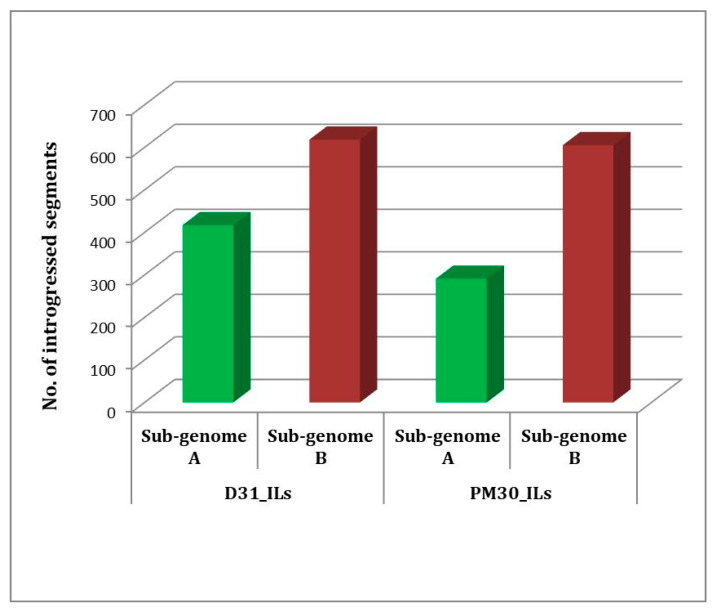
Introgressed segments of *B. carinata* into sub-genomes of *B. juncea* cultivars, viz., DRMRIJ 31 (D31_ILs) and Pusa Mustard 30 (PM30_ILs).

**Figure 5 plants-12-01677-f005:**
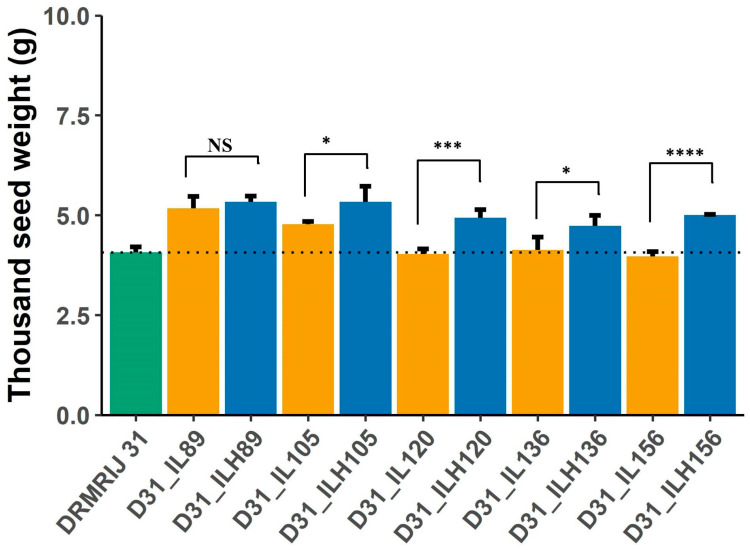
Mean performance of DRMRIJ 31, introgression lines in its genetic background (D31_ILs) and introgression line hybrids (D31_ILHs) for thousand seed weight (* significant at *p* = 0.1; *** significant at *p* = 0.01; **** significant at *p* = 0.001; NS = Non Significant).

**Figure 6 plants-12-01677-f006:**
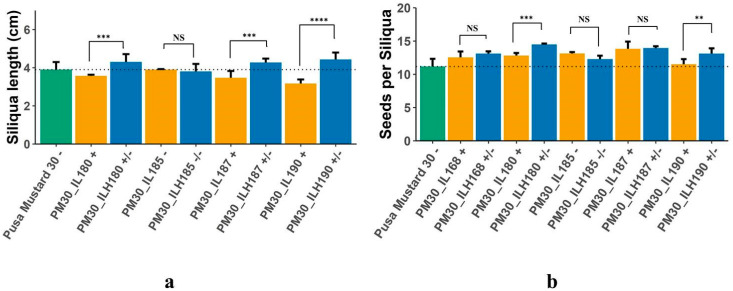
(**a**) Length of siliqua in Pusa Mustard 30, ILs of Pusa Mustard 30, and hybrids derived from them; (**b**) Number of seeds per siliqua in Pusa Mustard 30, ILs of Pusa Mustard 30, and hybrids derived from them (+ sign denote presence of *FLA3* gene in homozygous condition,—sign denote absence of *FLA3* gene, +/− sign denote heterozygosity of *FLA3* gene; ** significant at *p* = 0.05; *** significant at *p* = 0.01; **** significant at *p* = 0.001; NS = Non Significant).

**Table 1 plants-12-01677-t001:** Mean and range for seed yield and its contributing traits among genetic backgrounds, introgression lines (ILs) and introgression line hybrids (ILHs).

Genotype(s)	Parameter	Traits							
		SL	SPS	SMS	TS	OC	TSW	HI	SY
ILs in genetic background of DRMRIJ 31 (D31_ILs)	Mean ± SD	3.67 ± 0.51	12.7 ± 1.45	48.16 ± 4.36	307.20 ± 44.09	35.35 ± 2.39	4.52 ± 0.56	21.76 ± 2.66	2.05 ± 0.51
	Range	2.83–5.20	9.83–16.53	39.40–56.33	218–376.60	31.37–39.73	3.60–5.90	16.27–26.87	1.07–2.93
DRMRIJ 31	Mean	3.87	15.63	56.27	291.47	37.13	4.07	25.60	2.73
ILHs in genetic background of DRMRIJ 31 (D31_ILs × DRMRIJ 31)	Mean ± SD	4.02 ± 0.33	14.18 ± 0.86	51.03 ± 3.39	313.25 ± 40.27	36.37 ± 1.72	4.88 ± 0.33	24.8 ± 1.84	3.07 ± 0.39
	Range	3.27–4.73	12.73–16.23	44.73–56.67	237.93–387.67	32.10–39.50	4.30–5.63	20.03–28.60	2.40–4.07
Critical difference (*p* = 0.05)		0.401	1.4034	6.4279	68.7943	1.8523	0.6118	4.4766	0.4941
ILs in genetic background of Pusa Mustard 30 (PM30_ILs)	Mean ± SD	3.73 ± 0.46	13.47 ± 1.14	51.78 ± 6.42	357.12 ± 46.55	35.11 ± 1.77	4.81 ± 0.76	23.09 ± 2.37	2.38 ± 0.35
	Range	3.10–4.50	11.50–14.80	40.53–62.87	294.87–452.60	32.60–38.47	3.93–6.40	19.83–27.03	1.77–2.83
Pusa Mustard 30	Mean	3.90	11.17	43.87	233.80	31.77	5.97	21.30	2.03
ILHs in genetic background of Pusa Mustard 30 (PM30_ILs × Pusa Mustard 30)	Mean ± SD	4.21 ± 0.32	13.25 ± 0.70	49.53 ± 3.77	333.34 ± 51.52	35.31 ± 0.61	5.52 ± 0.49	24.54 ± 0.97	2.95 ± 0.31
	Range	3.80–4.70	12.07–14.47	44.20–55.07	267.80–434.20	34.07–36.17	5.07–7.00	22.27–26.10	2.47–3.47
Critical difference (*p* = 0.05)		0.4921	1.234	5.4801	68.6253	1.5683	0.6887	3.7609	0.4907

SL = Siliqua length (cm); SPS = Seeds per siliqua; SMS = Total siliquae on main shoot; TS = Total siliquae/plant; OC = Oil content (%); TSW = 1000 Seed weight; HI = Harvest index (%); SY = Seed yield (t/ha).

**Table 2 plants-12-01677-t002:** Extent of mid-parent heterosis (%) for seed yield and its contributing traits in introgression line hybrids (ILHs) generated from hybridizing *B. carinata*-derived *B. juncea* introgression lines with their respective genetic backgrounds.

Traits	ILHs in Genetic Background of DRMRIJ 31 (D31_ILs × DRMRIJ 31)		ILHs in Genetic Background of Pusa Mustard 30 (PM30_ILs × Pusa Mustard 30)	
	Mean	Range	Mean	Range
Siliqua length (cm)	7.79 ± 6.85	−6.33–19.40	10.56 ± 8.96	−4.84–25.47
Seeds/Siliqua	0.20 ± 5.69	−11.81–16.85	7.73 ± 6.20	−3.23–20.72
Total siliquae on main shoot	−2.73 ± 5.43	−11.67–8.67	3.81 ± 8.23	−8.23–16.98
Total number of siliquae/plant	5.16 ± 15.53	−17.67–33.95	14.01 ± 22.94	−16.92–51.34
Oil content (%)	0.00 ± 3.79	−11.93–6.57	5.65 ± 2.28	1.00–8.54
1000 seed weight (g)	13.48 ± 6.02	4.65–24.68	2.73 ± 9.23	−13.75–18.31
Harvest index (%)	5.00 ± 9.21	−21.41–24.12	10.82 ± 6.76	−1.10–22.92
Seed yield (t/ha)	31.36 ± 19.41	−7.69–76.81	34.19 ± 15.85	11.76–58.40

D31_ILs = *B. carinata*-derived *B. juncea* introgression lines in the genetic background of cultivar DRMRIJ 31. PM30_ILs = *B. carinata*-derived *B. juncea* introgression lines in the genetic background of cultivar Pusa Mustard 30.

**Table 3 plants-12-01677-t003:** Extent of standard heterosis (%) for seed yield and its contributing traits in test hybrids generated from hybridizing *B. carinata*-derived *B. juncea* introgression lines with a common tester (SEJ 8).

Traits	Hybrids between ILs of DRMRIJ 31 and SEJ 8 (D31_ILs × SEJ 8) ^#^		Hybrids between ILs of Pusa Mustard 30 and SEJ 8 (PM30_ILs × SEJ 8) ^$^	
	Mean	Range	Mean	Range
Siliqua length (cm)	−1.75 ± 8.40	−19.86–15.07	−0.47 ± 5.86	−9.09–11.36
Seeds/Siliqua	−2.05 ± 6.61	−13.97–12.23	9.02 ± 4.28	1.70–14.84
Total siliquae on main shoot	27.44 ± 6.79	10.6–41.47	9.25 ± 7.53	−2.68–22.43
Total number of siliquae/plant	33.96 ± 16.66	9.13–69.39	9.68 ± 11.07	−14.61–24.41
Oil content (%)	−2.28 ± 4.07	−11.13–4.07	−0.84 ± 2.77	−5.53–3.20
1000 seed weight (g)	−6.36 ± 7.03	−18.62–7.59	−14.2 ± 6.87	−26.03–−3.55
Harvest index (%)	6.09 ± 9.14	−8.55–25.91	11.64 ± 6.20	0.86–19.23
Seed yield (t/ha)	22.77 ± 13.11	−8.00–42.67	2.63 ± 11.97	−17.07–18.29

^#^ = Standard heterosis over check hybrid; DRMRIJ 31 × SEJ 8. ^$^ = Standard heterosis over check hybrid; Pusa Mustard 30 × SEJ 8.

**Table 4 plants-12-01677-t004:** Extent of heterosis (%) over mid-parent, respective introgression lines and genetic backgrounds for seed yield and its contributing traits in 10 Introgression Line Hybrids (ILHs).

Traits	Parameter	Introgression Line Hybrids (ILHs)									
		D31_ILH89	D31_ILH105	D31_ILH120	D31_ILH136	D31_ILH156	PM30_ILH168	PM30_ILH180	PM30_ILH185	PM30_ILH187	PM30_ILH190
SL	MPH	4.00	1.12	8.86	8.82	7.04	−1.29	15.18 *	−2.56	15.84 *	25.47 *
	Heterosis over IL	5.41	−11.76 *	4.88	23.33 *	15.15 *	−0.86	20.56 *	−2.56	23.08 *	40.00 *
	Heterosis over GB	2.63	18.42 *	13.16 *	−2.63	0	−1.71	10.26	−2.56	9.40	13.67 *
SPS	MPH	1.12	−3.43	3.45	−1.12	−4.32	10.55 *	20.72 *	0.96	11.47 *	15.59 *
	Heterosis over IL	21.62 *	−6.06	11.94 *	18.92 *	9.02	4.52	13.02 *	−6.60	0.72	13.91 *
	Heterosis over GB	−13.46 *	−0.64	−3.85	−15.38 *	−14.74 *	17.31*	29.55 *	9.85	24.78 *	17.31 *
SMS	MPH	−9.68	−3.36	3.33	0.54	−2.08	−4.81	11.78 *	15.08*	−8.23	−6.93
	Heterosis over IL	0.65	5.22	24.87 *	2.78	−0.18	−9.80	1.23	10.49	−18.51 *	−21.00 *
	Heterosis over GB	−18.09 *	−10.64	−11.88 *	−1.60	−3.90	0.76	24.77 *	20.06*	5.02	13.22 *
TS	MPH	4.27	−10.60	32.56 *	4.27	−9.17	12.95	30.70 *	0.13	−16.92	15.58
	Heterosis over IL	0.77	−16.41	35.36 *	−7.51	−14.05	−0.90	12.68	−20.08 *	−34.82 *	−0.47
	Heterosis over GB	8.03	−3.91	29.88 *	19.49	−3.70	31.31 *	55.57 *	34.02 *	14.54	37.81 *
OC	MPH	2.53	1.02	1.30	2.36	2.25	7.00 *	6.22 *	6.12 *	6.71 *	5.75 *
	Heterosis over IL	1.58	10.54 *	−2.01	5.73 *	7.40 *	1.51	−0.46	3.17	2.01	1.85
	Heterosis over GB	3.49	−6.99 *	4.84 *	−0.81	−2.42	13.12 *	13.85 *	9.23 *	11.86 *	9.97 *
TSW	MPH	16.13 *	20.45 *	20.99 *	15.66 *	23.46 *	7.79	4.60	−6.87	11.11	6.42
	Heterosis over IL	3.85	12.77	22.50 *	14.29	25.00 *	28.68 *	27.20 *	0	39.83 *	17.57 *
	Heterosis over GB	31.71 *	29.27 *	19.51 *	17.07 *	21.95 *	−7.26	−11.17	−12.85 *	−7.82	−2.79
HI	MPH	11.30	6.21	17.42	1.08	−4.00	15.55	9.79	22.92 *	−1.10	3.89
	Heterosis over IL	19.82	9.05	50.92 *	12.44	−6.32	19.10 *	1.05	23.31 *	−11.59	3.25
	Heterosis over GB	3.91	3.52	−3.91	−8.20	−1.56	12.21	20.19 *	22.54 *	12.21	4.54
SY	MPH	49.33 *	23.08 *	70.00 *	30.12 *	76.81 *	36.21 *	45.73 *	58.40 *	34.27 *	48.57 *
	Heterosis over IL	60.00 *	26.32 *	155.00 *	25.58 *	110.34 *	43.64 *	38.23 *	54.69 *	17.07 *	31.65 *
	Heterosis over GB	40.00 *	20.00 *	27.50 *	35.00 *	52.50 *	29.51 *	54.10 *	62.30 *	57.38 *	70.50 *

MPH, Mid-parent heterosis (%); IL, Introgression Line; GB, Respective genetic background; SL, Siliqua length (cm); SPS, Seeds per siliqua; SMS, Total siliquae on main shoot; TS, Total siliquae/plant; OC = Oil content (%); TSW, 1000 Seed weight (g); HI, Harvest index (%); SY, Seed yield (t/ha); * significant at *p* = 0.05.

## Data Availability

All the data are included within the manuscript and the Appendix A.

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
