# Peer review of "Introgression of Heterotic Genomic Segments from Brassica carinata into Brassica juncea for Enhancing Productivity"

_plants, 2023, doi:10.3390/plants12081677_

Round 1
Reviewer 1 Report
The manuscript is well written, even though, it needs some corrections (attached file). The methodology proposed by the authors is of interest for plant breeding, it is an alternative to existing methodologies for polyploidy detection. For this methodology to work with other species, previous knowledge is needed about the target sequences, that makes the methodology an alternative if the information is available.

Author Response
The manuscript is well written, even though, it needs some corrections (attached file). The methodology proposed by the authors is of interest for plant breeding, it is an alternative to existing methodologies for polyploidy detection. For this methodology to work with other species, previous knowledge is needed about the target sequences, that makes the methodology an alternative if the information is available.
Response: You have suggested a few correction as per attached file, however, the attached file is all together different manuscript. Please fetch the corrected version of our revised manuscript, which seems to be replaced by a different file.
Reviewer 2 Report
The authors; Vasisth et al. have addressed very good topic; Introgression of heterotic genomic segments from Brassica carinata into Brassica juncea for enhancing productivity. the present study is of great interest to develop the hybrids with higher seed and oil yield. Although, the manuscript contains good information but there are few flaws that may be addressed fulfilled for the validation of outcome.
Include all the parameters in the abstract.
Arrange the introduction section in systematic and scientific way with study gap and hypothesis.
It is better to to interpret the significant findings only rather than all/general information. Data presentation is poor, improve the quality of figure. Figure 1 is not clear.
Some figures and and tables are discussed in the discussion section, why not in results?
Material and methods section needs improvement; line 460, number of plants per row?
Conclusion must be short, It looks too lengthy even than the abstract. It may be short, specific and quantified.
Author Response
"Please see the attachment"

Reviewer 3 Report
Genetic resources utilization through introgression via interspecific hybridization is an efficient way to broaden genetic background and improve rapeseed productivity. The authors created introgression lines with B.Carinata and B. juncea. Further, they also developed IL hybrids and test hybrids and compared their heterosis. They also analyzed the segment effect of IL with different background and predicted many genes associated with seed yield and other traits. The result is exciting in the rapeseed genetic and breeding. However, I recommend rejection of this manuscript because there was only one-year field data. It is not reliable for yield and phenotypic changes with different years.
1. Are there any information on the statistic for those traits in Figure 1? Please re-structure the figure. Many redundant information can be removed.
2.Are there any information on the statistic for those traits in Table 1 the same as Table 2?
3.Please provide detailed information for Figure 4.
4. what's the meaning of Y aixs?
5. I suggest the authors remove Table 4 and 5 into supplementary data.
6.Please add Statistic information
Author Response
"Please see the attachment"

Reviewer 4 Report
The manuscript “Introgression of heterotic genomic segments from Brassica carinata into Brassica juncea for enhancing productivity” by Vasisth et al., describe the screen of ILs with significant mid-parent heterosis, and detected the introgressed segments using SNPs. I would recommend re-considering this manuscript after major revisions are provided.
Major revisions:
1.The author aligned sequenced reads from ILs to reference genome of B. juncea var. varuna UDSC Var 1.1, how the polymorphic SNPs between donor parent and respective genetic backgrounds were identified? In case that B. juncea is AABB genome while the donor is BBCC genome.
2. Fig 2, fig 3, fig 6 and fig 7 lack error bar, how many biological replication were used for each analysis?
3. what the 1:1.0, 2:2.0…………………..18:18.0 mean? What the legend A and B refer?
4 line 231-232, Introgression lines derived from the genetic background of DRMRIJ 31 carry 3.1% of the genome from B. carinata and 57.3% of it from B. juncea (DRMRIJ 31). What is the other 39.6%?
Minor revision
1. English need improve, such as “B. juncea is having a narrow gene pool…………” line 59, et
2. Line 98-101, discussion should not be placed in introduction
3. Line 225, “Five thousand one hundred and fifty seven” too long for 5150
4. The section conclusions should be placed following the discussions section.
Author Response
"Please see the attachment"

Round 2
Reviewer 3 Report
I am pleased that the authors has revised as I suggested. I can understand the difficulties for the authors to perform such big experiment for the data mining with two or more years.
Reviewer 4 Report
I have no more comments.